# Spatiotemporal Evolution of Residential Exposure to Green Space in Beijing

Yue Cao [1,2], Guangdong Li [1,2,*] and Yaohui Huang [1,2]

1   Key Laboratory of Regional Sustainable Development Modeling, Institute of Geographic Sciences and Natural Resources Research, Chinese Academy of Sciences, Beijing 100101, China
2   College of Resources and Environment, University of Chinese Academy of Sciences, Beijing 100049, China
*   Correspondence: ligd@igsnrr.ac.cn

**Abstract:** Urban green space has a critical impact on the urban ecological environment, residents' health, and urban sustainability. Quantifying residential exposure to green space and proposing targeted enhancement strategies in urban areas is helpful to rationally plan urban green space construction, reduce the inequality in residential exposure to green space, and promote environmental equity. However, the long-time evolution analysis of residential exposure to green space at different scales and the influence of green space quality on residential exposure to green space are rarely reported. Here we produced a long-time series dataset of urban green space from 1990 to 2020 based on the 30 m Landsat data and used the Normalized Difference Vegetation Index (*NDVI*) as a representation of the green space quality to comprehensively analyze residential exposure to green space at the city and block scales within the 5th ring of Beijing, China. We found that the urban green space in Beijing is mainly distributed in urban areas between the 4th and 5th rings (i.e., 153.4 km$^2$ in 2020), and there is little green space within the 2nd ring area (i.e., 12.6 km$^2$ in 2020). There is clear spatial inequality in residential exposure to green space, and about 2.88 million (i.e., ~27%) residents have experienced different degrees of decline in residential exposure to green space from 2015 to 2020. However, the degree of inequality in residential exposure to green space has gradually weakened from a high level (Palma ratio = 2.84) in 1990 to a relatively low level (Palma ratio = 0.81) in 2020. In addition, the spatial-temporal analysis method of residential exposure to green space based on green space quality has certain advantages that can help explore the degraded and lost areas of green space.

**Keywords:** urban green space; residential exposure to green space; green space quality; spatiotemporal evolution

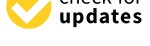



## 1. Introduction

The United Nations Sustainable Development Goal (SDG) 11 advocates sustainable urban development, improves the living environment of urban residents, and promotes the equality of urban green space (UGS) [1,2]. Rational UGS planning allows residents to enjoy relatively equal UGS at a low cost, improves residents' well-being, and enhances social cohesion. Over the past decades, the rapid urban expansion in China has greatly changed the original ecological pattern of urban areas. The urban structure and component units have also undergone obvious changes, and UGS is one of the vital units [3]. However, our understanding of the spatiotemporal evolution of UGS and inequality in residential exposure to green space is not clear enough to know whether urban residents are living in a "green enough" or "equally green" living environment. An in-depth understanding of the spatial distribution of UGS and the exploration of the spatiotemporal evolution of UGS inequality are necessary to improve UGS planning and enhance the equality of residential exposure to green space, thus reducing the negative environmental impact of urbanization and increasing the sustainability of urban development [4,5].

UGS is usually defined as areas with vegetation coverage in urban systems, including forests, grasslands, street trees, etc. [6]. UGS is important for improving the internal ecological environment of cities, such as mitigating the urban heat island effect and protecting biodiversity [7–12]. It is also an effective channel for urban residents to communicate with nature. For example, it can freshen the air, relieve residents' emotional stress, benefit residents' physical and mental health, and improve residents' happiness [13–17]. For UGS mapping, many studies use high-resolution images such as Quickbird and Worldview as data sources, which have rich spatial details and high accuracy for UGS extraction [18–20]. Unfortunately, the high cost and low processing efficiency make these images unsuitable for multi-time series and large-scale UGS mapping. Scholars prefer Landsat and Sentinel series images with open source and mature processing technology [21–23]. Furthermore, the continuous development and improvement of cloud platform processing technology in recent years has greatly improved the processing efficiency of these open-source remote sensing images. UGS datasets produced on cloud platforms such as Google Earth Engine (GEE) are emerging [21,24].

Green space accessibility is an effective means to measure the rationality of UGS distribution. It is derived from accessibility, which was first defined by Hansen as the potential for interaction opportunities [25]. Subsequently, it has been continuously developed and changed and has been widely used to measure the rationality of the layout of public resources such as UGS and entertainment facilities [26,27]. Green space accessibility refers to the degree of difficulty for urban residents to access nearby UGS [28]. It fully considers the spatiotemporal interaction between UGS and residents and reflects the ecological and social service functions of UGS. There are four main calculation methods to estimate green space accessibility. The first approach is the buffer analysis method, which reflects the distribution of UGS around residents by calculating the number or area of UGS in a certain buffer area where residents live [29,30]. The second approach is the nearest neighbor method, which only calculates the straight-line distance from the nearest UGS to residents [31]. The third approach is the gravity model, which considers the relationship between the attractiveness of the UGS itself and the competitiveness of people [32–36]. The fourth approach is the travel cost method, which refers to the calculation of the costs (such as money, time, etc.) spent by residents for the nearest UGS [37].

Nevertheless, the methods for calculating green space accessibility are not always perfect. The nearest neighbor method has a simple principle and high computational efficiency and is usually applied to the accessibility of specific types of UGS, such as parks. However, this method is very challenging to implement in urban environments with a large number of obstacles, such as buildings and water, and closer distance does not mean higher accessibility. The gravity model approach considers the intensity of UGS attracting residents to visit, which simulates the attraction of UGS to residents with a gravity model. The closer the distance and the smaller the number of people, the stronger the attraction. It reflects the competitive relationship between urban residents and public UGS resources and the degree of attraction of different characteristics and types of UGS to residents. However, individual differences are ignored, and the calculation method is relatively complicated, which is not conducive to large-scale and multi-time series research. The travel cost method is more consistent with the accessibility of traditional cognition, which is more in line with people's actual conception. It focuses on access at the physical level, considering the possible cost barriers such as time and financial resources for residents to go to UGS. However, this kind of visit usually lacks consideration of other socio-economic factors. Within a certain range, distance is not the key factor restricting UGS access. Furthermore, the complex calculation method also makes this method unfeasible in large-scale and multi-time series research. The buffer analysis method, that is, the residential exposure to green space method, can express the distribution of UGS in a certain area where residents are located and focuses more on the ecological benefits brought by UGS to residents, such as fresh air and a lower temperature. Unfortunately, such information as the type or quality of the UGS cannot be obtained [38,39].

Although existing studies have progressively considered the relationship between supply and demand for UGS, there are some shortcomings in the current research methodology for green space accessibility. The existing methods rarely consider the quality of UGS, yet the ecological and emotional benefits brought by different qualities of UGS to residents are somewhat different. Moreover, examining the spatial inequity of residential exposure to green space requires long time series and multi-scale analysis. Single time points and single scale data may not capture the dynamic change process of residential exposure to green space. To close these knowledge gaps, we produced a long time series dataset of UGS from 1990 to 2020 based on the 30 m Landsat data and used the *NDVI* as a representation of the UGS quality to comprehensively analyze the residential exposure to green space at the city scale and block scale within the 5th ring of Beijing. The contribution of this paper is that we comprehensively considered the quality and quantity of UGS in the assessment of green space accessibility. We proposed a new measurement method of residential exposure to green space based on *NDVI*, which can truly express the actual situation of UGS around residents. We also produced a long time series dataset of UGS from 1990 to 2020 within the 5th ring in Beijing based on the GEE platform, including seven phases (i.e., 1990, 1995, 2000, 2005, 2010, 2015, and 2020). We analyzed the residential exposure to green space at city scale and block scale in different periods. Then we evaluated the inequality in residential exposure to green space based on the Palma ratio to explore the regions and periods of unreasonable green space allocation.

## 2. Materials and Methods

### 2.1. Study Area

Beijing is the capital of China and one of the earliest cities in China to carry out urban UGS planning. As early as 1990, Beijing introduced the "Beijing Urban Greening Regulations", emphasizing the importance of UGS planning, and many related policies and regulations were also issued later. Although many studies have shown that with the rapid expansion in recent decades, the spatial pattern of UGS in the main urban area of Beijing has undergone major changes. Green space accessibility has shown significant inequality. Few studies have paid attention to the inequality of residential exposure to green space in different periods [40–42]. Here, we choose the built-up areas within the 5th ring, where a large population is distributed, as the study area (see Figure 1).

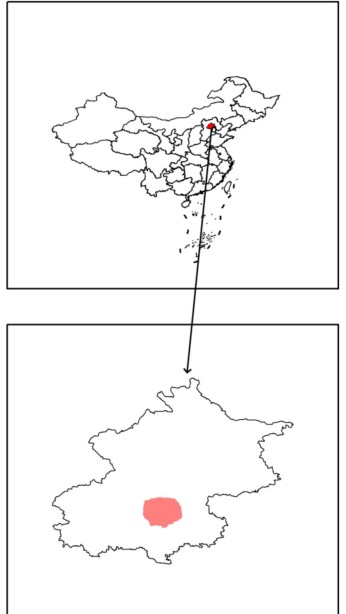
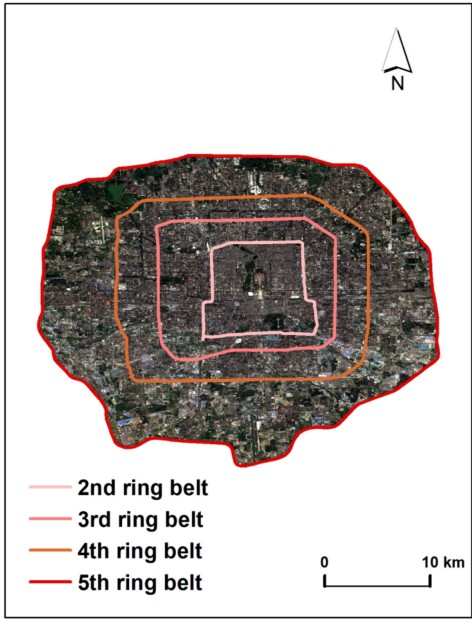

**Figure 1.** Study area.

### 2.1.1. Landsat Imagery

The Landsat satellite has been observing the Earth since 1972. The long-term data and relatively high resolution of 30 m make it the best choice for multi-time series research [43]. We used Landsat 4, 5, 7, and 8 surface reflectance data from June to August every year (i.e., 1990, 1995, 2000, 2005, 2010, 2015, and 2020) to extract UGS, which are open access and preprocessed on the GEE platform.

### 2.1.2. Urban Boundary Data

Since the study area is not always built up during the study period (the construction of the 5th ring was completed in 2003), we still need accurate boundary data of built-up areas to better quantify the residential exposure to green space. We used the Global Urban Boundaries (GUB) dataset, which is a global 30 m resolution urban boundary dataset based on the Global Artificial Impervious Area (GAIA) product and covering seven periods (i.e., 1990, 1995, 2000, 2005, 2010, 2015, and 2018), to represent the dynamics of the urban boundary in Beijing [44,45]. Then, we clipped the annual GUB data and finally obtained four years of urban boundary data (i.e., 1990, 1995, 2000, and 2005). In 2005, the interior of the 5th ring was all made up of built-up areas.

### *2.2. Data*

The data we used include remote sensing images, urban boundary data, population distribution data, and OpenStreetMap (OSM) data. See Figure 2 for the workflow.

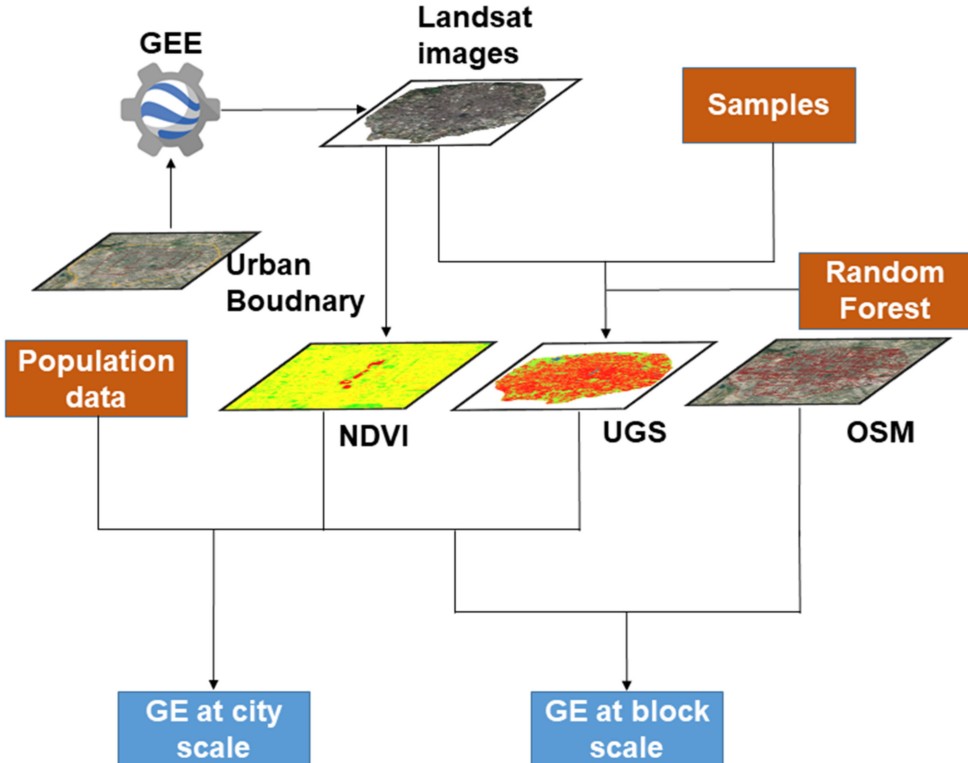

**Figure 2.** The flow chart. GEE—Google Earth Engine platform; *NDVI*—Normalized Difference Vegetation Index data; UGS—urban green space; OSM—OpenStreetMap data; *GE* at city scale is the residential exposure to green space at city scale and *GE* at block scale is the residential exposure to green space at block scale.

### 2.2.1. Population Data

We used the Global Human Settlement Population Grid (GHS-POP) dataset provided by the Joint Research Center (JRC) of the European Commission to represent the population

distribution information at the grid scale within the city. This dataset depicts the global population distribution at 5-year intervals from 1975 to 2030 at 100 m resolution [46].

### 2.2.2. OSM Data

OSM data are widely used in urban block division and functional plot extraction [47,48]. We chose the 2020 OSM data for block grid division. Based on this dataset, we explored the inequality of residential exposure to green space at the block scale. Due to the complexity of the block grid over time, that is, the 20-year block grid is subdivided on the basis of previous years, and the overall pattern has not changed evidently, it is feasible to only select the 2020 OSM data to divide the block grid.

### *2.3. Methods*

#### 2.3.1. Extraction of Urban Green Space

We used the random forest method to create the UGS dataset from 1990 to 2020 based on the multi-time-series Landsat dataset provided by the GEE platform. For each period, we filtered all images between 1 June and 1 September and synthesized them according to the maximum value of the *NDVI* to highlight the vegetation information as much as possible. The sample points in each period are obtained through visual interpretation and are divided into three categories: UGS, water, and built-up area. The number of sample points in each category ranges from 200 to 500 in each period. In order to better distinguish these three categories, we calculated the *NDVI*, Normalized Difference Built-up Index (NDBI), and Modified Normalized Difference Water Index (MNDWI) as auxiliary features for the random forest method input. Due to the complex internal environment of the city, the widespread dark pixels are often misclassified as water, which will cause the salt and pepper phenomenon, and the water extraction is not the main content of our research. Thus, we used the global surface water body dataset provided by JRC as a water mask [49].

#### 2.3.2. *NDVI* Based Residential Exposure to Green Space Assessment

We selected two scales (i.e., city scale and block scale) and three buffer radii (i.e., 100 m, 500 m, and 1000 m, respectively, representing visual distance, block distance, and maximum accessible distance) to quantify the residential exposure to green space in a specific grid [50,51].

Residential exposure to green space usually takes the form of the green coverage ratio (*GCR*), per capita green area, etc.

*GCR* is expressed as Equation (1):

$$GCR_i^\lambda = \frac{Green_i^\lambda}{Area_i^\lambda} \tag{1}$$

where $\lambda$ represents different buffer radii (i.e., 100 m, 500 m, and 1000 m), $Green_i^\lambda$ is the total area of UGS within the $\lambda$ range of the $i$th grid, and $Area_i^\lambda$ is the total area within the $\lambda$ range of the $i$th grid.

If population distribution data were taken into account [38,39], the residential exposure to green space (*GE*) is expressed as Equation (2):

$$GE^\lambda = \frac{\sum_{i=1}^N P_i \times GCR_i^\lambda}{\sum_{i=1}^N P_i} \tag{2}$$

where $P_i$ is the amount of the population in the $i$th grid, $GCR_i^\lambda$ is the green coverage ratio within the $\lambda$ range of the $i$th grid, and $N$ is the total number of grids in a specific unit.

However, the residential exposure to green space calculated by Equation (2) lacks information on the quality of UGS because it regards all green areas as homogeneous, which is contrary to the fact. The emotional benefits and ecological benefits brought to residents by low, sparse bushes and lush forests are very discrepant. Consequently, we use

*NDVI* to characterize the quality of UGS and combine it with population data to evaluate the residential exposure to green space, which can more truly express the distribution of UGS around residents. It can be calculated as:

$$GE\_w^\lambda = \frac{\sum_{i=1}^{N} P_i \times GCR_i^\lambda \times (NDVI_i^\lambda \times F)}{\sum_{i=1}^{N} P_i} \tag{3}$$

where $NDVI_i^\lambda$ is the mean value of *NDVI* within the $\lambda$ range of the *i*th grid, and *F* is the *NDVI* weight coefficient. The main consideration for the value of *F* is to make the UGS at the average *NDVI* level be the standard value (i.e., 1), so as to distinguish high-quality UGS from low-quality UGS. In this paper, *F* is taken as 1.8. So, the value of residential exposure to green space is the average proportion of standard-quality green space in a certain buffer zone around residents in a special unit. Its value is a comprehensive reflection of the level of UGS around residents. A low value of residential exposure to green space is not conducive to the physical and mental health of residents, while a high value will cause a problem of excess resources. Moreover, how to choose an appropriate value depends on the actual situation, taking into account socio-economic factors such as the economic development of different countries and cities. Ensuring a relatively equal exposure to green space for residents is beneficial for sustainable urban development, and this is a challenge that managers need to address.

### 2.3.3. Inequality in Residential Exposure to Green Space

Due to the unreasonable distribution of UGS, residential exposure to green space varies in different periods and regions, and the opportunities for residents to access UGS are not always equal. The Palma ratio can be used to assess this inequality in residential exposure to green space [52]. In previous studies, the Gini coefficient was frequently used to measure the inequality in residential exposure to green space. However, it was too sensitive to the moderately exposed people and could not fully express the inequality between the low-exposure people and the high-exposure people [53–56]. As a substitute for the Gini coefficient, the Palma ratio can explore the inequality between the top 10% of exposed people and the bottom 40% of exposed people in a certain area. The Palma ratio is calculated as follows:

$$\text{Palma} = \frac{\sum_{i=N \times 90\%}^{N} GE_i}{\sum_{i=1}^{N \times 40\%} GE_i} \tag{4}$$

where *N* is the total population, and the residential exposure to green space is ranked from low to high; $N \times 40\%$ represents the exposure of the 40th percentile residents, $N \times 90\%$ represents the exposure of the 90th percentile residents. $GE_i$ is the residential exposure to green space of the *i*th resident.

## 3. Results

### *3.1. Urban Green Space Map*

#### 3.1.1. Accuracy Assessment

We obtained 150 validation sample points in 2015 and 2020, respectively, to examine the performance of UGS extraction based on the visual interpretation of Google Images. The results showed that the classification accuracy was relatively high, and the overall accuracy was 0.91 and 0.90, respectively. We found that our results are very consistent with those of Google Images (Figure 3). We can vaguely see the road outline in Figure 3, which reflects the accuracy of our extraction of roadside UGS. Our UGS map not only contains large areas of UGS but also effectively identifies some small patches of UGS. Meanwhile, the results of our extraction of the UGS inside the blocks are also satisfactory, which provides a solid foundation for our following research.

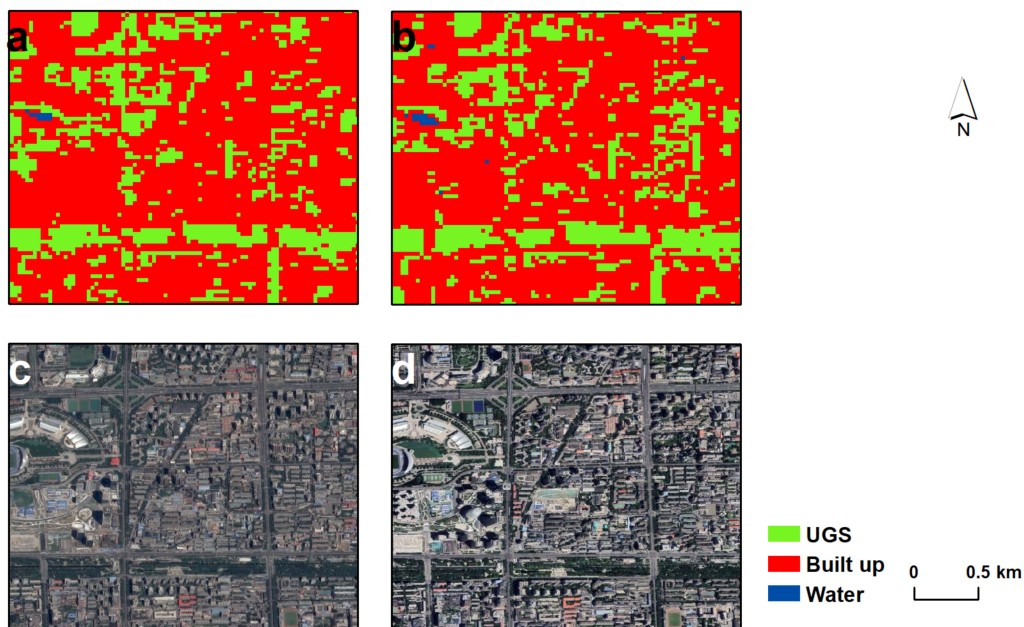

**Figure 3.** An example of accuracy assessment. (**a**) Urban green space map in 2020; (**b**) Urban green space map in 2015; (**c**) Google Image on August 3, 2020; (**d**) Google Image on 4 August 2015. UGS—urban green space.

3.1.2. Spatial-Temporal Pattern of Urban Green Space in Beijing

The distribution of UGS in Beijing from 1990 to 2020 is shown in Figure 4. We found that the green area gradually increases from the city center to the periphery. There is little UGS inside the 2nd ring. The total amount of UGS between the 4th and 5th rings is the largest, and it is also the area with the most drastic changes. The UGS has gradually changed from the initial uniform distribution in the four directions to the north-south distribution, and the amount of UGS in the east and west of Beijing has slowly decreased. The green coverage ratio within the 5th ring presents a trend of first decreasing and then slowly rising (Table 1). The turning point occurred in 2005. The green coverage ratio increased from 0.295 in 2005 to 0.325 in 2010, and the UGS increased by about 20 km$^2$. This is consistent with the previous results [57].

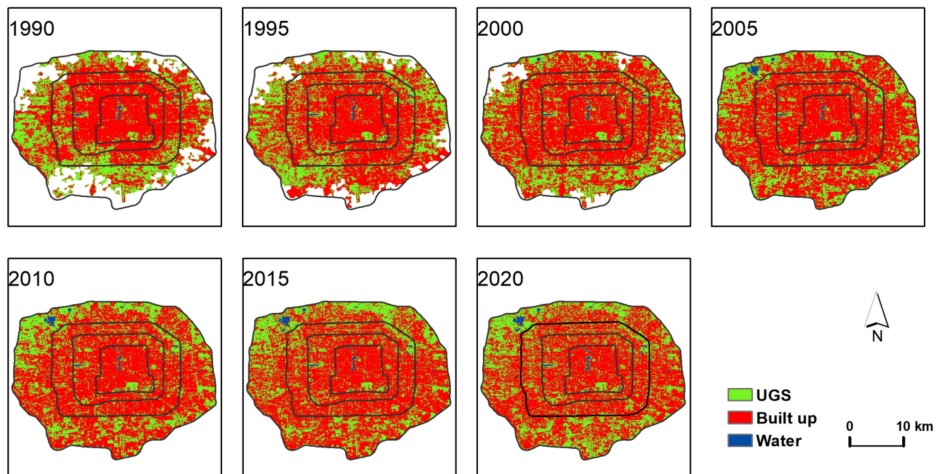

**Figure 4.** Spatial pattern of green space in Beijing from 1990 to 2020. UGS—urban green space.

**Table 1.** Distribution quantity and coverage rate of urban green space between rings. *GCR*—green coverage ratio; Ring 2—the area within the 2nd ring; Ring 2_3—the area between the 2nd ring and the 3rd ring; Ring 3_4—the area between the 3rd ring and the 4th ring; Ring 4_5—the area between the 4th ring and the 5th ring.

| Year | GCR | Ring 2 | | Ring 2_3 | | Ring 3_4 | | Ring 4_5 | |
|------|-----|--------|--------|----------|--------|----------|--------|----------|--------|
| | | Area (km$^2$) | Percent (%) | Area (km$^2$) | Percent (%) | Area (km$^2$) | Percent (%) | Area (km$^2$) | Percent (%) |
| 1990 | 0.353 | 7.1 | 11.3 | 15.3 | 15.9 | 45.7 | 31.8 | 125.7 | 51.7 |
| 1995 | 0.345 | 8.6 | 13.7 | 20.2 | 21.0 | 42.2 | 29.3 | 134.9 | 46.2 |
| 2000 | 0.312 | 10.4 | 16.6 | 19.3 | 20.0 | 30.3 | 21.1 | 137.8 | 41.9 |
| 2005 | 0.295 | 8.9 | 14.2 | 18.0 | 18.7 | 29.2 | 20.3 | 141.0 | 38.6 |
| 2010 | 0.325 | 10.6 | 16.9 | 20.9 | 21.7 | 35.8 | 24.9 | 149.7 | 40.9 |
| 2015 | 0.331 | 10.4 | 16.6 | 22.8 | 23.7 | 39.6 | 27.5 | 148.2 | 40.5 |
| 2020 | 0.348 | 12.6 | 20.1 | 25.1 | 26.1 | 41.9 | 29.1 | 153.4 | 42.0 |

Table 1 shows the amount and coverage of UGS among the rings from 1990 to 2020. Since the 5th ring was not constructed before 2000, only the green coverage in the built-up area is calculated. We found that the area and proportion of UGS are gradually increasing from the 2nd ring to the 5th ring. Most of the UGS is distributed in areas between the 4th and 5th rings, away from the city center, and the inside of the 2nd ring is only about 10 km$^2$. The UGS is growing overall; each ring exhibits different trends. In the past 30 years, the UGS inside the 2nd ring has increased by ~78%, and the UGS between the 2nd ring and 3rd ring has increased by 64%, while the UGS between the 3rd ring and 4th ring has slightly decreased (~9%). The area of UGS between the 4th ring and 5th ring continued to increase, but the proportion of UGS began to rise slowly until the urbanization of the 5th ring road was basically completed in 2005 due to the urban expansion. It is worth noting that the change trend of UGS between the 3rd and 4th rings is highly consistent with the green coverage ratio. During 1990–2005, the UGS loss in this area was one of the main reasons for the overall green coverage ratio decline. During 2005–2020, the UGS gain in this area was an important factor in the increase of the green coverage ratio, which indicates that this area is a worthwhile area for studying the structure change of the UGS in Beijing.

*3.2. Residential Exposure to Green Space at Different Scales*

3.2.1. City Scale

From 1990 to 2020, the green coverage ratio showed a downward trend and then a continuous upward trend, and 2005 was an inflection point. This may be due to the rapid urban expansion, which led to a large growth of the city before the 5th ring road was fully completed. Yet the corresponding increase in UGS was less or even negative in some areas (as shown in Table 1, the UGS loss is about 11.9 km$^2$ between the 3rd and 4th rings from 1995 to 2000). After 2005, although human activities inside the 5th ring were still intense, the loss of UGS was small. The government had built many green spaces in preparation for the 2008 Olympic Games, which led to a substantial increase in green coverage ratio in 2010. However, contrary to expectations, the residential exposure to green space at the city scale shows a multi-stage trend that is not always consistent with the green coverage ratio, which reflects, to some extent, the balance between the supply and demand of UGS. Figure 5 shows that although the green coverage ratio in 1995 declined compared to 1990, the residential exposure to green space increased substantially, which indicates that the UGS structure and quality were relatively better in 1995 and the limited UGS met the requirements of more residents. A well-organized UGS structure and high quality can improve the residential exposure to green space, but there is a limit to this improvement. The sharp decline in the green coverage ratio in 2000 led to a large reduction in residential exposure to green space this year. From 2010 to 2020, the green coverage ratio increased slightly, but the residential exposure to green space increased drastically, which shows that

a greater amount of UGS and a reasonable distribution of UGS have a synergistic effect. We found that residential exposure to green space shows an upward trend from 1990 to 2020, not only because of the increase in the total amount of UGS but also because of a more reasonable distribution of UGS. In other words, to make UGS more accessible to more people, it is necessary to provide more UGS and a reasonable distribution of UGS.

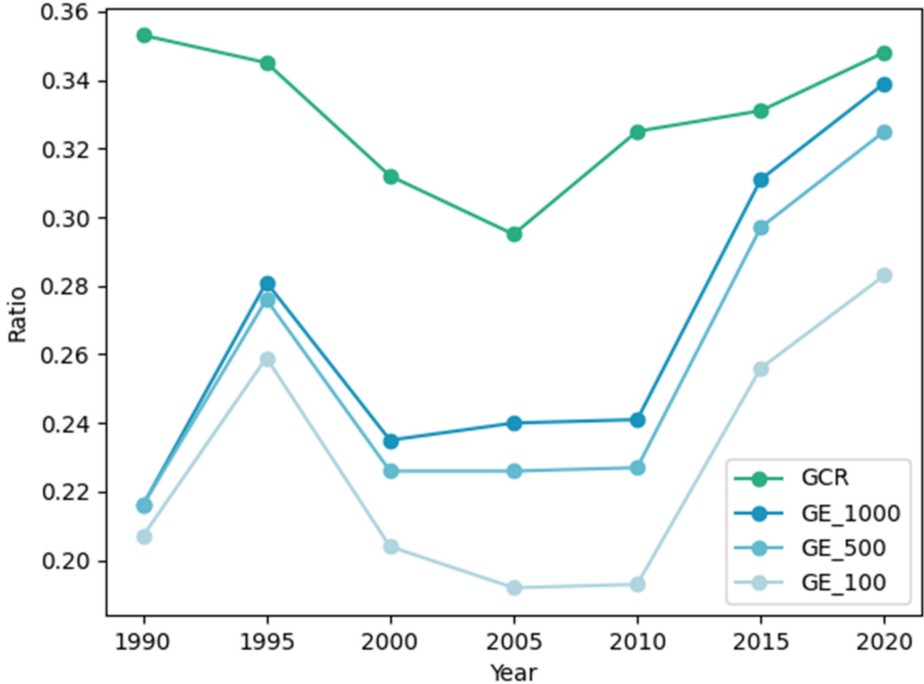

**Figure 5.** Green coverage ratio and the residential exposure to green space in different buffer radii at city scale. *GCR*—green coverage ratio; *GE_*1000—the residential exposure to green space in 1000 m buffer radius at city scale; *GE_*500—the residential exposure to green space in 500 m buffer radius at city scale; *GE_*100—the residential exposure to green space in 100 m buffer radius at city scale.

Moreover, we found that with the expansion of the buffer radius (100 m-500 m-1000 m), the residential exposure to green space will increase, which indicates that most luxuriant UGS is far from the residents and there is little green space in the community. This difference can be directly expressed by the residential exposure to green space difference in different buffer radii. Figure 6 shows that the residential exposure to green space difference between 1000 m and 500 m in each time period is considerably smaller than the difference between 500 m and 100 m, and the difference slowly increases over time. That is, within a 1000-m range centered on residents, the green space coverage in the 100-m ring is obviously less than that outside the 100-m ring, and the UGS in the residents' visual distance is few. Furthermore, the irrationality of the UGS distribution is gradually increasing, which deserves our attention.

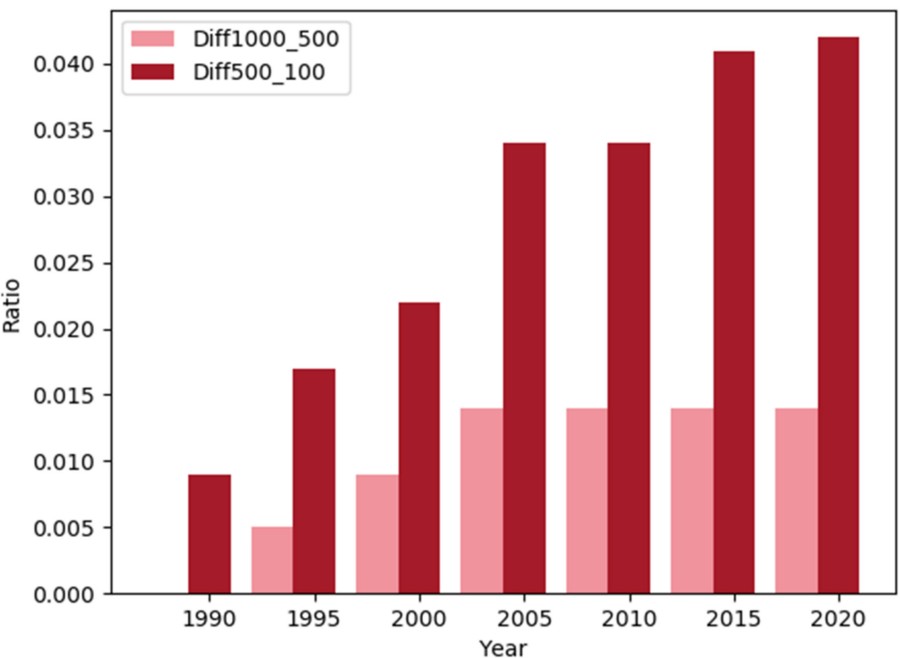

**Figure 6.** Differences in the residential exposure to green space of different buffer radii at city scale. Diff1000_500—the difference between the residential exposure to green space in 1000 m buffer radius and the exposure in 500 m buffer radius; Diff500_100—the difference between the residential exposure to green space in 500 m buffer radius and the exposure in 100 m buffer radius.

### 3.2.2. Block Scale

We used the 2020 OSM data for block grid division and divided the residential exposure to green space in each block into 7 levels. Figure 7 shows the distribution of residential exposure to green space at block scale from 1990 to 2020 (the buffer radius is 500 m). We found that residential exposure to green space at the block scale has obvious spatiotemporal heterogeneity. The residential exposure to green space shows an increasing trend from the inner to the outer city. The inner blocks of the 2nd ring are clear low-value (<0.3) concentration areas. The residential exposure to green space of the blocks between the 4th ring and the 5th ring has been at a high level (>0.4), which mainly contributes to the high green space coverage near the 5th ring road.

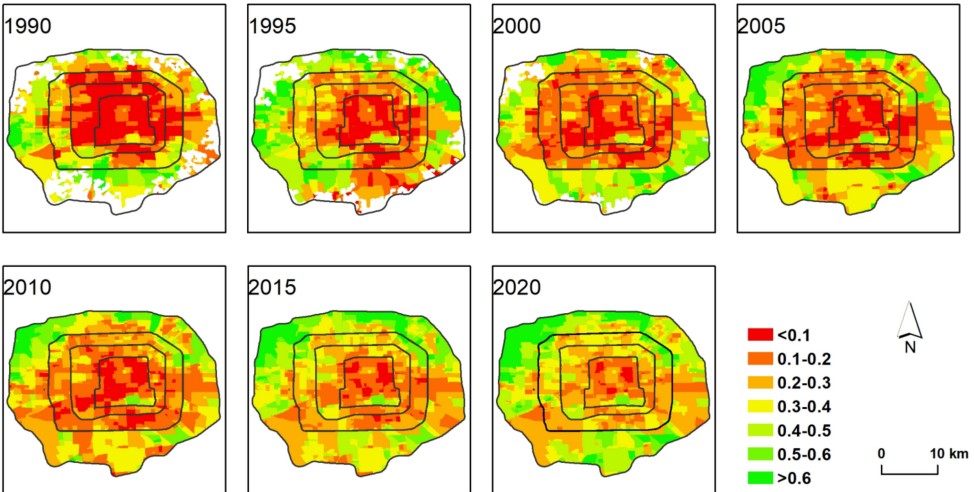

**Figure 7.** Spatial pattern of the residential exposure to green space at block scale.

Moreover, the residential exposure to green space of 26.8% of the blocks was at a high level in 1995 (see Table 2), and more than 1.5 million people live in areas with lush green space, which was clearly higher than that before and after and confirmed the rationality of the UGS structure in 1995. Most residents (i.e., more than 70%) were at a low exposure level between 2000 and 2010, and the low-value areas of residential exposure to green space were not reduced until 2015. Meanwhile, the high-value areas of residential exposure to green space have also expanded, and the blocks have been "greening". In summary, with the promotion of UGS construction, residential exposure to green space has improved to some extent, the low-value areas of residential exposure to green space in the central urban area are shrinking, and the opportunities for residents to access UGS have increased.

**Table 2.** Statistics of the exposure level of the blocks every year. High is the number of blocks with a high level of exposure (i.e., the residential exposure to green space > 0.4), and Low is the number of blocks with a low level of exposure (i.e., the residential exposure to green space < 0.3). Population_High and Population_Low are the total number of people in these blocks, respectively. Total_Block is the total number of blocks, and Total_Population is the total population within the 5th ring in Beijing every year.

| Year | High | Percent (%) | Population _High | Low | Percent (%) | Population _Low | Total _Block | Total _Population |
|------|------|-------------|------------------|-----|-------------|-----------------|--------------|-------------------|
| 1990 | 196 | 18.0 | 805,481 | 739 | 68.0 | 3,248,869 | 1086 | 4,602,075 |
| 1995 | 296 | 26.8 | 1,544,493 | 669 | 60.7 | 3,434,704 | 1103 | 5,603,949 |
| 2000 | 139 | 12.4 | 894,805 | 845 | 75.2 | 5,068,238 | 1123 | 6,704,684 |
| 2005 | 142 | 12.5 | 887,286 | 861 | 76.0 | 5,959,260 | 1133 | 8,130,211 |
| 2010 | 122 | 10.8 | 939,829 | 857 | 75.6 | 7,273,041 | 1133 | 9,726,315 |
| 2015 | 272 | 24.0 | 2,516,773 | 651 | 57.5 | 5,895,015 | 1133 | 10,271,718 |
| 2020 | 274 | 24.2 | 2,810,004 | 590 | 52.1 | 5,608,351 | 1133 | 10,727,545 |

However, not all blocks have improved the residential exposure to green space after 2015, and the residential exposure to green space for 30.5% of the blocks is gradually reduced from 2015 to 2020. This means that about 2.88 million people are in an environment that is getting worse. In fact, due to urban expansion and regeneration, some blocks have already experienced the degradation and loss of UGS, which are widely present in new urban areas and the urban village on the edge of the 5th ring. For example, there was a large area of trees in the north of the block in Figure 8 in 2015, but the trees degenerated into shrubs and grassland in 2020, and the UGS became a built-up area in 2023. The degradation and loss of UGS have resulted in a substantial decrease (i.e., from 0.326 to 0.229) in residential exposure to green space in the block and adjacent blocks, severely limiting residents' access to UGS and reducing the well-being of residents in these blocks. The degradation and loss of UGS need to be addressed by relevant government departments, and new UGS should be built in other areas to compensate for the degradation and loss of UGS. Meanwhile, the decline of residential exposure to green space in 2020 also illustrates the advantages of the residential exposure to green space method based on *NDVI* weighting. The traditional residential exposure to green space method regards the detected UGS as homogeneous, which is not sensitive enough to the degradation of UGS. Under this assumption, the residential exposure to green space in the blocks does not change much. Using the *NDVI* to characterize the quality of UGS can accurately detect the degraded area of UGS so as to better characterize the real condition and distribution of UGS around residents.

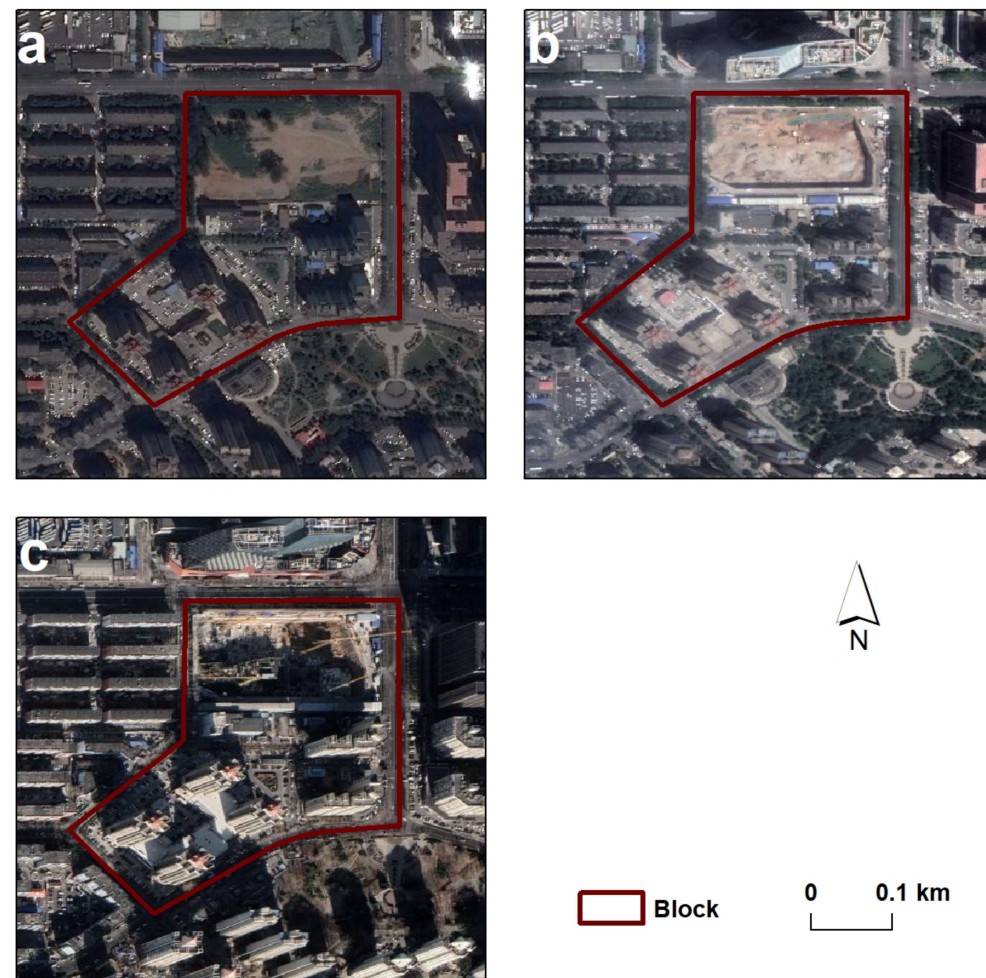

**Figure 8.** An example of green space degradation and loss. The base map comes from Google Images, and the time of picture is (**a**) 11 July 2015, (**b**) 15 July 2020, and (**c**) 3 February 2023.

### 3.2.3. Exploration of Influencing Factors

According to Equation (3), we can find that green space and population jointly determine the size of the residential exposure to green space. There have been obvious changes in both green space and population with the development of the city. Consequently, we have explored the impact of these two factors on residential exposure to green space. Here, population reflects the utilization rate of green space. Within a certain range, the more population there is in a particular green space, the higher the utilization rate of the green space, and thus the higher the residential exposure to green space. As the population density differences are small at the block level, the residential exposure to green space depends more on the distribution of green space in the block. Therefore, we analyzed the impact of population and green space distribution on residential exposure to green space at the city level. We found that the population within the 5th ring in Beijing has been continuously increasing and approaching saturation from 1990 to 2020, but the residential exposure to green space did not show this trend (see Table 3). In fact, population growth leads to changes in population distribution, which in turn affects residential exposure to green space. We can assume that the population is evenly distributed (i.e., the utilization rate of all green space is consistent) when we do not consider population factors, and thus we can obtain a new value, GE_green. The uneven distribution of the population results in a GE_green that is about 20% higher than the corresponding *GE* in the same year. This indicates that areas with high green coverage have low population densities and low utilization of green space. In the city's core area, although the population density is high, there is little green space.

**Table 3.** Changes in population and residential exposure to green space. *GE*—the residential exposure to green space with population weighting; GE_green—the residential exposure to green space without population weighting; and r—the Pearson's correlation coefficient ($p < 0.01$).

| Year | Population | *GE* | GE_green | r |
|------|-----------|------|----------|---|
| 1990 | 4,602,075 | 0.216 | 0.264 | −0.293 |
| 1995 | 5,603,949 | 0.276 | 0.334 | −0.272 |
| 2000 | 6,704,684 | 0.226 | 0.300 | −0.328 |
| 2005 | 8,130,211 | 0.226 | 0.310 | −0.354 |
| 2010 | 9,726,315 | 0.227 | 0.295 | −0.330 |
| 2015 | 10,271,718 | 0.297 | 0.358 | −0.271 |
| 2020 | 10,727,545 | 0.325 | 0.391 | −0.310 |

Meanwhile, we calculated the similarity between green space distribution (using the green coverage rate) and population distribution and used Pearson's r to represent this correlation (see Table 3). Theoretically, a positive correlation between green space and population distribution would lead to a significant improvement in residential exposure to green space. However, here we observed that for all years, there was a weak negative correlation between green space and population distribution, and the difference is not obvious from year to year. Areas with high green coverage often come with low population densities and low green utilization rates. In fact, this is a common phenomenon within cities, where high population density areas often mean high costs for green space construction. Building more green space in areas with low population density has little effect on improving resident well-being. Green space construction should not blindly pursue quantity but should be people-oriented and serve a wider range of populations. Balancing the cost of green space and population distribution is the core issue of urban green space construction.

*3.3. The Inequality of Residential Exposure to Green Space*

Figure 9 shows that a clear spatial heterogeneity in the residential exposure to green space across different blocks, and this spatial differentiation varies with time. Residents in different blocks have different possibilities for accessing UGS. Therefore, we use the Palma ratio to quantify the degree of inequality in the distribution of residential exposure to green space. When the Palma ratio is 1, it means that the average residential exposure to green space in the high-exposure group is four times that of the low-exposure group. Figure 9 shows that with the gradual increase of residential exposure to green space at the city scale, the degree of inequality in residential exposure to green space at the block scale gradually weakens, but it is still at a high level. The inequality in residential exposure to green space at the block scale within the 5th ring of Beijing has changed from a high value (i.e., 2.84 in 1990) to a relatively low value (i.e., 0.81 in 2020). The changes in the Palma ratio imply that the average residential exposure to green space for the high exposure group versus the low exposure group decreases from 11.36 times in 1990 to 3.2 times in 2020. We also found that the central blocks of Beijing were mainly low-value areas of residential exposure to green space in 1990, which was in sharp contrast with the high-value of the blocks between the 4th ring and 5th ring (Figure 7). In contrast, although blocks with low residential exposure to green space still exist, their number has decreased, and the area with a low exposure value has shrunk. Consequently, the difference in residential exposure to green space between blocks has decreased considerably in 2020.

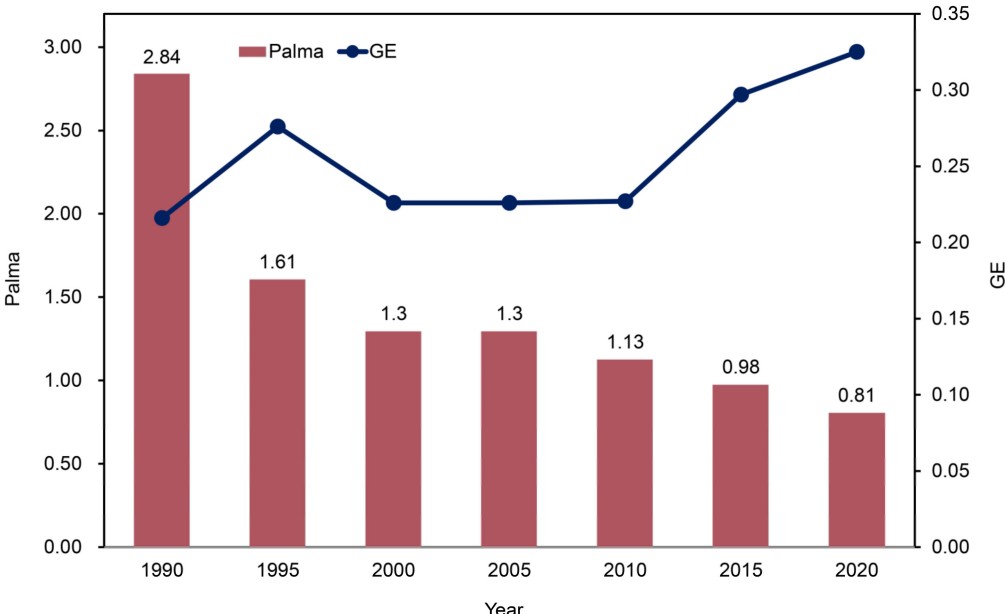

**Figure 9.** Inequality assessment of residential exposure to green space at block scale. *GE*—the residential exposure to green space at city scale; Palma—the Palma ratio. When the Palma ratio is 1, it means that the average residential exposure to green space in the high-exposure group is four times that of the low-exposure group.

Overall, the degree of inequalities in residential exposure to green space has gradually weakened over time, and the opportunities for residents in different blocks to access UGS are becoming relatively equal. This implies that Beijing's UGS construction has achieved remarkable achievements in the past 30 years. However, there is still room for improvement. For example, the residents living in the inner 2nd ring are still experiencing a terrible green environment, and about 2.88 million people are in an environment that is getting worse after 2015, according to our research. In the future, UGS planning needs high-quality construction, and we should focus on the areas where there is little UGS (such as the inner area of the 2nd ring) or the UGS degrades and disappears due to urban renewal, thus mitigating the inequality in the residential exposure to green space and promoting environmental justice.

## 4. Discussion

This study produced a multi-time-series UGS dataset in areas within the 5th ring of Beijing, analyzed the spatiotemporal evolution of residential exposure to green space, and evaluated the degree of the inequality in residential exposure to green space using the Palma ratio over the past 30 years. It provides a scientific method to support the construction of UGS in Beijing. Compared with previous studies, we have the following contributions [27,30,39]. First, previous studies on green space accessibility and green space inequality assumed that all UGS is homogeneous and the benefits that UGS bring to residents are consistent. However, this is clearly contrary to reality. Sparse street flower beds and dense gardens are both UGS, but they are quite different in terms of ecological benefits and emotional benefits to residents [58]. The emotional benefits of gardens to residents are obviously higher than those of street flower beds. Therefore, we use *NDVI* as a characterization of UGS quality to distinguish high-quality UGS, such as tall trees, from low-quality UGS, such as sparse grassland, to more truly express the distribution of UGS around residents. Our method can accurately identify areas of degradation and loss of UGS and make the research more reasonable.

Second, we considered the spatiotemporal evolution of residential exposure to green space in Beijing, which is rarely involved in previous studies [58–61]. We can interpret the

residential exposure of green space from a multidimensional perspective and explore the low-value areas and areas that are getting worse. Moreover, we can identify areas that have been of low value, such as the inner area of the 2nd ring in Beijing, to carry out targeted construction and reduce inequality in residential exposure to green space.

However, our research still needs to be improved. Here, we used the *NDVI* maximum synthesis method to minimize the impact of building shadows on the extraction of urban green space. The method can highlight the vegetation information to its maximum extent and improve the extraction accuracy. Nevertheless, the effect of this method may be limited in the case of dense buildings and long shadows. Moreover, the *NDVI* value can only represent the lushness of UGS; that is, it lacks consideration of the overall landscape pattern of UGS, such as the different types of forests, grasslands, and street trees [58]. In addition, there is a lack of consideration of social factors, such as the difference between park UGS and private community UGS [62]. Although the *NDVI* values of these different types of UGS are similar, they have a great impact on the assessment of residential exposure to green space, which is an aspect that needs to be considered in future research. Furthermore, our research area needs to be expanded [30,39]. In the future, we can consider residential exposure to green space on a national or even global scale and analyze its spatiotemporal evolution and inequality evaluation under a unified framework to help achieve sustainable development.

## 5. Conclusions

In this study, we produced a long time series dataset of UGS from 1990 to 2020 in Beijing based on the 30 m Landsat data, which has more than 90% accuracy. We used the *NDVI* weighted method to comprehensively analyze the spatiotemporal evolution of residential exposure to green space at the city scale and block scale within the 5th ring in Beijing from 1990 to 2020. We then evaluated the degree of inequality in residential exposure to green space so as to provide scientific method support and a decision-making reference for the UGS planning in Beijing. We found that the *NDVI* weighted method can address the lack of UGS quality in assessments of residential exposure to green space and can accurately identify areas of degradation and loss of UGS. It expresses the distribution of UGS around residents more realistically and enriches the theoretical research on green space accessibility. The distribution of UGS and residential exposure to green space in Beijing has clear spatial differentiation. Although they show a certain upward trend, some blocks with about 2.88 million people still experienced degradation and loss of UGS from 2015 to 2020. Here, we observed that there was a weak negative correlation (i.e., ~0.3, $p < 0.01$) between green space and population distribution. Areas with high green coverage may have low population densities and low green utilization rates. Green space construction should not blindly pursue quantity but should be people-oriented and serve a wider range of populations. Moreover, residents in Beijing have experienced a certain degree of inequality in terms of residential exposure to green space (Palma ratio = 0.81 in 2020), especially the inner 2nd ring area. The total amount of UGS and the residential exposure to green space have always been at an extremely low level inside the 2nd ring, which needs to be paid attention to in future UGS construction.

**Author Contributions:** Conceptualization, Y.C.; methodology, G.L.; software, Y.C.; validation, Y.C., Y.H.; formal analysis, G.L.; investigation, G.L.; resources, G.L.; data curation, Y.C.; writing—original draft preparation, Y.C.; writing—review and editing, Y.C. and G.L.; visualization, Y.C.; supervision, G.L.; project administration, G.L.; funding acquisition, G.L. All authors have read and agreed to the published version of the manuscript.

**Funding:** This research was funded by the National Natural Science Foundation of China (Grant No. 41971207), the Second Tibetan Plateau Scientific Expedition and Research Program (STEP) (Grant No. 2019QZKK1005), and the Youth Innovation Promotion Association CAS (Grant No. 2020053).

**Data Availability Statement:** The data presented in this study are available on request from the author.

**Conflicts of Interest:** The authors declare no conflict of interest.

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
