# Peer review of "Spatiotemporal Evolution of Residential Exposure to Green Space in Beijing"

_remotesensing, doi:10.3390/rs15061549_

Round 1
Reviewer 1 Report
1. The evaluation method has weighted the population data, but this element has not appeared in the subsequent quantitative and qualitative analysis. The population information should be added.
2. Section 3.3.2 of the paper illustrates the advantages of the evaluation method based on NDVI: it can detect the reduction of the residential exposure to green space caused by the degradation of green space, but the magnitude of this reduction is not explained. This should be explained clearly. In addition, this part can be listed in a separate section, and other supporting evidence should be added.
3. The authors should explain exactly what the residential exposure to green space means, especially in section 3.2, and need to be clear about what it means in practice.
4. The authors should add some recent literatures in the discussion section.
5. The study area of this manuscript is the area within the 5-ring road of Beijing, and larger scales can be considered in the future, which can be explained in the discussion section.
Author Response
非常感谢您的评论。我们已经在文件中进行了回复(见附件)。再次感谢。

Reviewer 2 Report
In the manuscript titled "Spatiotemporal evolution of residential exposure to green space in Beijing", the authors mainly used the time series Landsat images to analyze the residential exposure to green space in Beijing, and the methods and results were clearly described. The conclusion is solid. However, there is also some weakness to need to modify. Here are some comments for authors:
1) The authors emphasized that the quality of UGS was very important which was rarely considered in line 102. But I do not find how the authors evaluated the quality of UGS in this manuscript. Just using NDVI? I think it's not enough. I suggested that classifying the green space types such as trees, shrubs, or grass may be more suitable. So in eq (3), the NDVI coefficient was not suitable.
2) It is very common that the shadow of high buildings influenced the green space extraction in the city. In results 3.1, the authors showed the UGS maps were very well. So, how the authors considered the shadows in this manuscript?
3) With the development of the city, the green space and population will both change. I think the change in population and the green space are very sensitive to residential exposure to green space. So, I suggested that you should also analyze the changes in the population in the city and block level and find out how these two factors influenced the GE.
Based on these reasons, I suggested the manuscript needs major revision.
Author Response
Thank you very much for your comments. We have responded to them in the document (Please see the attachment). Thanks again.

Reviewer 3 Report
This is an interesting study and I’m sure that if developed, can be used in the future. However, many affirmation in the study need to be corrected, as the authors themselves point out at the end the Discussion chapter, there are many factors not taken into account in their analysis. For this reason I suggest a review of the entire text with attention to the quality and attractivity assessments - see the use of the NDVI Weight method with only specific (and few) factors.
Below you can see some of the notes I took while reading your paper:
- review the text for there are some minor English issues.
- you have a series of obvious statements as well as some that are very vague throughout the text.
- line 84 – is it fewer?
- I don’t agree with the phrase from line 331-334 as this affirmation cannot be result of only this analysis. There are other reasons for which a space is attractive to a group of people: neighbourhood, services in the area (shops, museum, etc …), schools, family/friends, etc. Just because you identified a preference for some UGSs, this does not make them “high-quality”. You must support this affirmation.
- line 338-340 again an inaccurate conclusion if you don’t provide more information – for instance, the dimensions of the UGSs found within 100 m and 1000 m from the residential area. Obviously that 1000 m one might have more options. The UGS found at 100 m from a residential area (again, what is the size and density – typology - of the residential area) might be of small size, overcrowded … there are so many factors to consider in order to state that a space has quality of not.
- 344-352 – flawed affirmation – Two points: first, as the rings are concentric, as you go from the interior towards the exterior the surface in discussion increases and second, there is the issue of density. The 2nd and 5th ring have different patterns of urban tissue and obviously there will a larger quantity of UGSs between the 4th and 5th rings – as mentioned before, quality is a different thing, and your study does not support this affirmation.
- refrain from qualitative statements unless you conduct a qualitative study. You talk generally about UGS without any classification of them: gardens, squares, urban parks, zoo, etc.; type of vegetation; presence of body of water in the UGS; services in the area; leisure; etc.
402-403 – compare the 5th ring to the others (numbers), not only to itself, for a more accurate result. The 5th ring developed and densified later and thus it was expected to have a UGS drop. Also, you are missing the state of the art regarding UGs ratios – give examples of good practices, UGS ratios in other cities (countries), exposure considered good in literature.
- line 410 – I doubt the size of the block shrunk … exposure increased due to new UGSs, I assume.
- you cannot state in line 415 that UGSs are becoming relatively equal and then in lines 417-418 say that in the 2nd ring there is high inequality.
- line 418-419 – in UGS planning should always aim for high-quality, no? Or only in the center of Beijing?
Author Response

(The authors gave the same response as above.)

Round 2
Reviewer 1 Report
Accept in present form.
Reviewer 3 Report
I am happy with the author's response.